# Effects of Extraction Methods on the Physicochemical Properties and Biological Activities of Polysaccharides from *Polygonatum sibiricum*

**DOI:** 10.3390/foods12102088

**Published:** 2023-05-22

**Authors:** Yongshuai Jing, Meng Yan, Hao Zhang, Dongbo Liu, Xiaoyue Qiu, Beibei Hu, Danshen Zhang, Yuguang Zheng, Lanfang Wu

**Affiliations:** 1College of Chemistry and Pharmaceutical Engineering, Hebei University of Science and Technology, 26 Yuxiang Street, Shijiazhuang 050018, China; cjys1985@126.com (Y.J.);; 2College of Pharmacy, Hebei University of Chinese Medicine, 3 Xingyuan Road, Shijiazhuang 050200, China

**Keywords:** *Polygonatum sibiricum*, polysaccharides, extraction methods, physicochemical properties, biological activity

## Abstract

*Polygonatum sibiricum* polysaccharides (PSPs) have important biological functions, such as antioxidation, immunomodulatory, and hypolipidemic functions. Different extraction methods have effects on their structures and activities. In this study, six extraction methods, including hot water extraction (HWE), alkali extraction (AAE), ultrasound-assisted extraction (UAE), enzyme-assisted extraction (EAE), microwave-assisted extraction (MAE), and freeze–thaw-assisted extraction (FAE) were used to extract PSPs, and their structure–activity relationships were analyzed. The results showed that all six PSPs had similar functional group compositions, thermal stability, and glycosidic bond compositions. PSP-As (PSPs extracted by AAE) exhibited better rheological properties due to their higher molecular weight (Mw). PSP-Es (PSPs extracted by EAE) and PSP-Fs (PSPs extracted by FAE) had better lipid-lowering activity due to their lower Mw. PSP-Es and PSP-Ms (PSPs extracted by MAE), which do not contain uronic acid and have a moderate Mw, had better 1,1-diphenyl-2-picrylhydrazyl (DPPH)-radical-scavenging activity. On the contrary, PSP-Hs (PSPs extracted by HWE) and PSP-Fs, with the Mw of uronic acid, had the best OH-radical-scavenging activity. The high-Mw PSP-As had the best Fe^2+^-chelating ability. In addition, mannose (Man) may play an important role in the immunomodulatory activity. These results indicate that different extraction methods affect the structure and biological activity of polysaccharides to varying degrees, and these results are helpful for understanding the structure–activity relationship of PSPs.

## 1. Introduction

*Polygonatum* mainly includes *Polygonatum sibiricum* Red, *Polygonatum kingianum* Coll. et Hemsl, and *Polygonatum cyrtonema* Hua. More than 60 species of *Polygonatum* have been reported in the world, and there are about 30 species in China, which are mainly distributed in Yunnan, Sichuan, Guizhou, Guangxi, and other places in China, and also in Myanmar and Vietnam [1]. As a traditional Chinese medicine, the rhizomes of these species have the functions of invigorating the spleen, moistening the lung, tonifying the kidney, promoting blood circulation, and enhancing immunity [2]. As a medicinal and food resource, *Polygonatum* has a broad application value. The main functional and active components of *Polygonatum* are polysaccharides, flavonoids, lignins, steroidal saponins, anthraquinones, amino acids, alkaloids, etc. [3]. Polysaccharides are one of the main components of *Polygonatum*, and they have antioxidant, hypolipidemic, hypoglycemic, immunity-regulating, and other effects [4,5].

Some scholars have found that the polysaccharides obtained are different depending on the extraction rate, structure, and activity method due to the influence of extraction process factors such as the temperature, the solid–liquid ratio, and the extraction medium [6,7]. The extraction methods of polysaccharides typically include hot water extraction (HWE), alkaline extraction (AAE), ultrasonic-assisted extraction (UAE), compound-enzyme-assisted extraction (EAE), microwave-assisted extraction (MAE), and freeze–thaw-assisted extraction (FAE) [8,9]. Many reports have shown that polysaccharides obtained by different extraction processes have significant differences in their physicochemical properties that affect their antioxidant, hypolipidemic, and immunomodulatory activities [10,11]. In the 2020 edition of Chinese Pharmacopoeia, polysaccharides are a quality control index for *Polygonatum*. *Polygonatum sibiricum* polysaccharides (PSPs) have also been widely examined because of their functional activity and edible value. Whether different extraction methods affect the structure and biological activity of PSPs needs to be further explored. Therefore, it is important to evaluate and compare the chemical structures and biological activities of PSPs obtained by different extraction processes for the application of PSPs and the exploration of their structure–activity relationships.

Plant polysaccharides are diverse and complex, so we used six different methods to extract PSPs. Ultraviolet–visible spectroscopy (UV-vis), Fourier-transform infrared spectroscopy (FT-IR), high-performance liquid chromatography (HPLC), high-performance gel permeation chromatography (HPGPC), thermogravimetric analysis (TGA), nuclear magnetic resonance (NMR), and scanning electron microscopy (SEM) were used to evaluate and compare the structural characteristics of PSPs extracted by different extraction processes. The rheological properties, antioxidants, and lipid-lowering and immunomodulatory activities were further analyzed. The purpose of this study was to investigate the influence of different extraction processes on PSPs and to provide a reference for the practical application and development of PSPs.

## 2. Materials and Methods

### 2.1. Experimental Materials and Reagents

*Polygonatum sibiricum* Red (*P. sibiricum*) (No. 2022050101) was obtained from Anguo Yaoyuan Trading Co., Ltd. (Anguo, China). The sample was identified as the root of *P. sibiricum* by associate Professor Lanfang Wu (Department of Pharmacy, Hebei University of Chinese Medicine). A voucher specimen was deposited at the College of Chemistry and Pharmaceutical Engineering, Hebei University of Science and Technology, China. 1,1-Diphenyl-2-picrylhydrazyl (DPPH), standard monosaccharides, 1-phenyl-3-methyl-5-pyrazolone (PMP), salicylic acid, and phenanthroline were purchased from Aladdin Biochemical Technology Co., Ltd. (Shanghai, China). The citric acid, phenol, concentrated sulfuric acid, chloroform, and Cell Counting Kit-8 (CCK-8) were provided by Tianjin Yongda Chemical Reagent Co., Ltd. (Tianjin, China). Trifluoroacetic acid (TFA) was provided by Sigma-Aldrich (Shanghai) Co., Ltd. (Shanghai, China). Acetonitrile and methanol were provided by Merck (Germany). H_2_O_2_ and FeCl_2_ were provided by Tianjin Damao Chemical Reagent Factory (Tianjin, China). Orlistat, sodium glycocholate hydrate, sodium taurocholate, and olive oil were provided by Shanghai Yuanye Biotechnology Co., Ltd. (Shanghai, China). RAW264.7 macrophages were provided by Shanghai Cell Bank Center, Chinese Academy of Sciences (Shanghai, China). Methyl sulfoxide (DMSO) cellulase, papain, and acetic acid were provided by Beijing Soleibo Technology Co., Ltd. (Beijing, China). Naphthalene ethylenediamine hydrochloride and p-amino benzenesulfonamide were provided by Shanghai Linen Technology Development Co., Ltd. (Shanghai, China). All chemicals used were of analytical grade.

### 2.2. Extraction Methods

#### 2.2.1. Pretreatment Process

The roots of *P. sibiricum* were cut into pieces with scissors, then cold-soaked in 95% ethanol (1:35, *w*/*v*) for 24 h at 25 °C. They were taken out, allowed to dry, and ground into powder using a pulverizer (600Y, Yongkang Platinum Europe Hardware Products Co., LTD, Jinhua, China).

#### 2.2.2. Hot Water Extraction

The method of Yuan et al. [7] was referred to. Briefly, the pretreated *P. sibiricum* powder was extracted 2 times with hot water (1:30, *w*/*v*) at 85 °C for 2 h. After being concentrated, it was precipitated with 95% ethanol (1:4). The samples were let stand at 4 °C for 12 h, centrifuged at 4000× *g* rpm for 10 min (TGL-15B, Shanghai Anting Scientific Instrument, Shanghai, China), and filtered under reduced pressure (120 mm, Brinell’s funnel); the precipitate was then collected and dried [12]. The PSPs were obtained and named PSP-Hs.

#### 2.2.3. Alkali-Assisted Extraction

The pretreated *P. sibiricum* powder was added to a certain volume of 3% NaOH (1:30, *w*/*v*) and soaked at 25 °C for 2 h. The alkaline leach solution was neutralized to a pH of about 7 and filtered under reduced pressure. After being concentrated, it was precipitated with 95% ethanol (1:4). The samples were let stand at 4 °C for 12 h, centrifuged at 4000× *g* rpm for 10 min, and filtered under reduced pressure (120 mm, Brinell’s funnel); the precipitate was then collected and dried [13]. The PSPs were obtained and named PSP-As.

#### 2.2.4. Ultrasonic-Assisted Extraction

The pretreated *P. sibiricum* powder was added to an ultrasonic instrument (60 °C water, 100 W, KS-300DE, Kunshan Jielimei Ultrasonic Instrument Co., Ltd., Kunshan City, China) at a ratio of 1:30, *w*/*v*, subjected to ultrasonic extraction for 40 min, and heated in a water bath at 80 °C for 80 min, for a total of 2 times. The extracts were pooled, concentrated, and precipitated in alcohol for 12 h at 4 °C; centrifuged at 4000× *g* rpm for 10 min; and filtered under reduced pressure (120 mm, Brinell’s funnel). The precipitate was then collected and dried [14]. The PSPs were obtained and named PSP-Us.

#### 2.2.5. Enzyme-Assisted Extraction

The pretreated *P. sibiricum* powder, along with 2% of the sample weight of cellulase and papain, was added to water (1:30, *w*/*v*) and activated at 40 °C for 10 min. The pH was adjusted to about 5.5, enzymolysis was performed for 2 h, the solution was boiled for 10 min to inactivate it, and the filtrate was combined and filtered. After being concentrated, it was precipitated with 95% ethanol (1:4). The samples were let stand at 4 °C for 12 h, centrifuged at 4000× *g* rpm for 10 min, and filtered under reduced pressure (120 mm, Brinell’s funnel); the precipitate was then collected and dried [15]. The PSPs were obtained and named PSP-Es.

#### 2.2.6. Microwave-Assisted Extraction

The pretreated *P. sibiricum* powder was added to water (1:30, *w*/*v*). Microwave (700 W, Midea, Guangdong Midea Kitchen Appliance Manufacturing Co., Ltd., Foshan, China) extraction was performed for 20 min, and the samples were then placed in a water bath for 40 min. The above steps were performed 1 time, and the solution was extracted twice. After being concentrated, it was precipitated with 95% ethanol (1:4). The samples were let stand at 4 °C for 12 h, centrifuged at 4000× *g* rpm for 10 min, and filtered under reduced pressure (120 mm, Brinell’s funnel); the precipitate was then collected and dried [16]. The PSPs were obtained and named PSP-Ms.

#### 2.2.7. Freeze–Thaw-Assisted Extraction

The pretreated *P. sibiricum* powder and water (1:30, *w*/*v*) were frozen at −80 °C for 12 h and thawed for 2 h at 65 °C. The extraction was performed 2 times. After being concentrated, the solution was precipitated with 95% ethanol (1:4). The samples were let stand at 4 °C for 12 h, centrifuged at 4000× *g* rpm for 10 min, and filtered under reduced pressure (120 mm, Brinell’s funnel); the precipitate was then collected and dried [16]. The PSPs were obtained and named PSP-Fs.

### 2.3. Characterization of PSPs

#### 2.3.1. UV-Vis Spectra and FT-IR Spectra Analyses of PSPs

The total sugar content and protein content were determined according to established laboratory methods [17]. In brief, the PSPs were dissolved in water (0.1 mg/mL) and their total sugar content was measured by the phenol–sulfuric acid method using UV-vis (L5S, Shanghai INEIdian Analytical Instrument, Shanghai, China) to measure the absorbance at 490 nm. The PSPs (0.1 mg/mL) were scanned with UV-vis (L5S, Shanghai INEIdian Analytical Instrument, China) at the full wavelength from 200 to 800 nm to measure the protein content. PSPs powder and KBr particles were evenly mixed and ground into a fine powder, and an appropriate amount of mixed powder was pressed into a transparent, uniform, thin sheet. FT-IR (S-100, PerkinElmer, Waltham, MA, USA) scanning was performed in the wavelength range of 4000~500 cm^−1^ to identify the functional group structure of each PSPs [18].

#### 2.3.2. Analysis of Monosaccharide Composition and Average Molecular Weight (Mw) of PSPs

Amounts of 3 mL of 2.0 mol/L TFA and 10.0 mg of PSPs were mixed and hydrolyzed at 110 °C for 6 h. After hydrolysis, the mixture was co-distilled with MeOH and derivatized with PMP; then, the pH was adjusted to neutral and samples were obtained by chloroform extraction. The monosaccharide analysis system consisted of an Agilent 1260 HPLC system and an ODS C18 column (4.6 mm × 250 mm). The sample was eluted with phosphate buffer (pH of 6.8) and acetonitrile in a ratio of 17:83; the flow rate was 1 mL/min and the column temperature was 25 °C. The monosaccharide standard was determined under the same conditions as above [19]. The proportion of monosaccharides was calculated according to Equation (1). The Mw was determined according to a previously established method [19]:(1)$$\mathrm{Proportion}\;\mathrm{of}\;\mathrm{monosaccharides}\;\%\;=\frac{A_{1}}{A_{0}}\times 100$$
where A_0_ represents the total peak area and A_1_ indicates the peak area of the monosaccharide component.

#### 2.3.3. TGA of PSPs

The TGA was based on our previous study [20]. It was conducted on a synchronous thermal analyzer (Q600, New Castle, DE, USA). The PSPs were measured under a N_2_ environment at a heat rate of 10 °C/min and a temperature range of 25~800 °C.

#### 2.3.4. SEM Analysis of PSPs

The PSPs were fixed on the sample stage with conductive tape, operated at a 10 kV acceleration after gold spraying, and the samples were analyzed by SEM (Noran system 7, JEOL, Akishima shi, Japan) [21].

#### 2.3.5. NMR Analysis of PSPs

The samples were separately dissolved in 0.5 mL of DMSO, and then ^1^H NMR was recorded on an AVANCE NEO 500 MHz spectrometer (Bruker, Mannheim, Germany) with a 5 mm CPTCI ^1^H probe [22].

#### 2.3.6. Rheological Property Analysis

Existing studies were referred to and slightly modified [23]. PSPs solutions (4, 8, and 16 mg/mL) were rheometrically tested using a rheometer (HAAKE MARS 40, Karlsruhe, Germany). The steady-state shear viscosity was maintained at shear rates from 1 to 100 s^−1^. Small-amplitude oscillatory shear experiments were performed at 25 °C to generate shear storage (G′) and loss (G″) moduli at frequency ranges of 1~100 rad/s [24,25]. The effect of temperature was also measured. When the sample concentration was 4 mg/mL, the steady-state shear viscosity and dynamic rheological properties of the samples were determined at different temperatures (20, 60, and 100 °C).

### 2.4. Evaluation of Biological Properties of PSPs

#### 2.4.1. In Vitro Antioxidant Activity

Metabolites such as reactive oxygen species (ROS) and free radicals are produced in the process of oxidation. Free radicals can react with proteins, lipids, DNA, and other substances; destroy their structure; and have adverse effects on the body’s normal metabolism. Therefore, the antioxidant activities of PSPs extracted by different processes were compared. Based on the work of others, the free-radical-scavenging activities (DPPH and OH) and the chelating ability for Fe^2+^ were determined [26,27]. Vitamin c (Vc) and EDTA were used as positive controls, respectively, to evaluate their activity by IC_50_.

#### 2.4.2. In Vitro Hypolipidemic Activity

The method of Li et al. [28] was slightly modified. An amount of 0.2 mL of PSPs solution and 1.0 mL of an olive oil emulsifier (0.229 g/mL) were added to 1 mL of phosphate buffer (pH of 7.4). After 10 min in a water bath at 37 °C, a pancreatic lipase solution was added, and then after 15 min in the water bath, ethanol (3.0 mL) and a 1% phenolamine indicator were added. A NaOH solution (0.1 M) was added and the titration volume was recorded as A_1_. Distilled water was used instead of the sample solution for the blank control. The volume was titrated and recorded as A_0_. Orlistat was used as a positive control. The inhibition rate of pancreatic lipase by PSPs was calculated according to Equation (2):(2)$$\mathrm{Rate}\;\mathrm{of}\;\mathrm{inhibition}\;\%\;=\frac{A_{0}-A_{1}}{A_{0}}\times 100$$
where A_0_ is the titration volume of the blank control group and A_1_ is the titration volume of the sample group.

Different concentrations of PSPs solutions were mixed with a pepsin solution (1.5 mL, 10 mg/mL) and HCl (0.5 mL, 0.01 M) and placed in a 37 °C water bath for 1 h. A trypsin solution (2 mL, 10 mg/mL) was added. Then, a glycocholate solution (2 mL, 0.4 mM) and a taurocholate solution (2 mL, 0.5 mM) were added and the mixture was incubated at 37 °C for 1 h. A cholate solution was added and the mixture was then incubated for 1 h at 37 °C. The absorbance of its supernatant at 387 nm was measured, and its content was calculated using a standard curve (glycocholate: y = 2.8138x + 0.027, R^2^ = 0.9994; taurocholate: y = 2.4076x + 0.0113, R^2^ = 0.9976) [29].

#### 2.4.3. In Vitro Immunomodulatory Activity

The effect of PSPs on macrophage viability was studied by the CCK-8 method. After adjusting the concentration to 2 × 10^4^ cells/mL, the cells were inoculated in 96-well plates; each well contained a volume of 100 μL and was cultured in a cell incubator. After 24 h, 100 μL of PSPs (10~500 μg/mL) at different concentrations were used to replace the culture medium, and lipopolysaccharide (LPS) (1 μg/mL) was added to the positive control group. After 20 h, 0.5 mg/mL CCK-8 was added to the 96-well plate. After 4 h of continuous culture, the supernatant was removed and DMSO (150 μL) was added to the wells. The wells were shaken and mixed evenly to fully dissolve the purple methyl methane crystals. The absorbance was measured by a microplate reader at 490 nm [12].

The RAW264.7 cells were incubated for 6 h and then incubated with different concentrations of PSPs (10~500 μg/mL) or 1 μg/mL LPS for 24 h. The Griess reagent was used to determine the content of nitric oxide (NO). The absorbance at 540 nm was measured by a microplate reader [30,31]. The standard curve for the NO content was y = 0.0112x + 0.0297, R^2^ = 0.9991.

### 2.5. Statistical Analysis

All experimental data are shown as means ± SD. A one-way analysis of variance (ANOVA) plus a post hoc Duncan’s test (SPSS software, version 26.0) was used to evaluate the statistical significance. For all the results among the different groups, *p* < 0.05 was considered a significant difference.

## 3. Results and Analysis

### 3.1. Characterization

#### 3.1.1. Compositions and Extraction Rates of PSPs

The results showed that the extraction rate and total soluble sugar content of PSPs were closely related to the extraction method. As shown in Table 1, the PSP-Hs extraction rate was the highest (14.86%). It was thus speculated that polysaccharides are more soluble at high temperatures. The extraction rate of polysaccharides was improved by reducing the pull and viscosity.

In addition, according to the standard curve of the total sugar content of Glc (y = 15.673x − 0.0153, R^2^ = 0.9996), the total soluble sugar content in PSPs was between 76.61 and 87.14%, indicating that polysaccharides were the main component of each sample. Figure 1a shows that the PSPs all contained a small amount of protein.

#### 3.1.2. Functional Group Composition of PSPs

The structural features of PSPs were determined by FT-IR. The FT-IR spectra of PSPs prepared by different extraction processes are shown in Figure 1b and Table 2. The peaks at 3276 cm^−1^ and 2930 cm^−1^ were caused by the OH stretching vibration and the C-H asymmetric stretching vibration, indicating that all six PSPs had the characteristic absorption peaks of polysaccharides [32]. There were three main peaks near 1629, 1414, and 1014 cm^−1^. The results show that the signal near 1603 cm^−1^ was attributed to the stretching vibration of the vibration absorption peak caused by O-H/C-O/COO-. The 1414 cm^−1^ peak may be related to the C-H and CH_2_ bending vibration absorption peaks. The absorption peak at 1014 cm^−1^ indicates various vibrations of the C-O-C glycosidic bond and the C-O-H bond [18,33]. It is noteworthy that the range of 1020~1050 cm^−1^ was also attributed to S=O and C-O stretching, while the absorption peaks of PSP-As, PSP-Us, PSP-Ms, and PSP-Hs were relatively strong here. Therefore, it is speculated that this is related to the better gel properties of PSP-As, PSP-Us, PSP-Ms, and PSP-Hs. Similar studies have also shown that these absorption peaks have an important effect on the gel formation and emulsification ability, or the ability to interact with water to generate hydrogen bonds, thereby forming suitable gels and emulsions [33]. In addition, the stretching vibration of PSPs at around 875 cm^−1^ indicated that there were polysaccharides containing β-cyclic glycosidic bonds. The absorption peaks at approximately 875 cm^−1^ and 813 cm^−1^ were composed of Man residues [32]. In conclusion, different extraction processes did not significantly affect the primary structure of PSPs.

#### 3.1.3. Analysis of Monosaccharide Composition and Average Mw of PSPs

The monosaccharide composition of PSPs was analyzed, and the obtained chromatograms are shown in Figure 1c. Table 3 below shows the monosaccharide ratio and composition of each monosaccharide. According to the results, the monosaccharide composition of PSPs extracted by the six methods differed; Man and Glc were the main monosaccharides in PSPs. The difference was that PSP-Us, PSP-Es, and PSP-Ms did not contain galacturonic acid (GalA) and galactose (Gal), and the other three polysaccharides contained a small amount of GalA and Gal. PSP-Ms contained more xylose (Xyl) than the PSPs of other extraction methods. Moreover, the PSP-As and PSP-Us treatments caused the hydrolysis of polysaccharide chains and the breakage of intermolecular hydrogen bonds, thereby affecting the monosaccharide composition. According to the HPGPC spectrum (Figure 1d), relevant information can be obtained. The Mw was obtained from the standard curve (y = −0.3108x + 11.749, R^2^ = 0.9911). The Mw of the six PSPs was compared as follows: PSP-As (5964 Da) > PSP-Us (5726 Da) > PSP-Ms (5466 Da) > PSP-Hs (4025 Da) > PSP-Es (3926 Da) > PSP-Fs (3563 Da) [19].

#### 3.1.4. Thermal Stability Analysis of PSPs

The thermal stability of the six PSPs was investigated using a TGA. As shown in Figure 1e, the six samples had a similar thermal stability, and all had three different weightlessness phases. The first stage occurred between room temperature and 220 °C, mainly due to the disappearance of free water. With increasing temperature, the polysaccharide mass decreased rapidly between 220 and 380 °C. According to previous studies, 220~380 °C is the temperature range for polysaccharide degradation [34]. In addition, when the temperature was increased to 800 °C, most of the PSPs were carbonized into ash and inorganic components. This high-temperature degradation indicates that the PSPs had a good thermal stability and were very stable over a wide temperature range [35].

#### 3.1.5. SEM Analysis of PSPs

The microstructure of PSPs was studied using SEM. As shown in Figure 2, the surface morphology of the PSP-Hs was relatively complete. The PSP-As had some holes, but the distribution of the holes was not uniform. The surface of the PSP-Us was void, rough, uneven, and had a block-like wrinkle structure. The PSP-Es showed irregular block distribution. The surface of the PSP-Ms has unevenly distributed pores. The surface of the PSP-Fs was rough, uneven, and had a wrinkled structure with deep holes. Both the PSP-Us and PSP-Fs had wrinkled structures, which may have been due to the high energy of the ultrasound and the ultra-low temperature of −80 °C, which split the structure of PSPs into small molecular fragments [36].

#### 3.1.6. NMR Analysis of PSPs

The ^1^H-NMR chromatograms of PSPs are shown in Figure 1f. The hydrogen signals are shown in Table 4. The pyranose configuration was judged from a chemical shift value of δ 4.95 ppm [37,38]. It can be found that there were pyranose configurations linked by α and β configurations in the PSPs. In addition, the six PSPs all contained an anomeric hydrogen signal with a chemical shift of δ 5.20 ppm. Combined with the research results of Tao [39], it is speculated that the Glc residue is in the α-type configuration.

#### 3.1.7. Steady Shear Flow Behavior

The rheological flow behavior of polysaccharides was closely related to their Mw distribution and molecular chain entanglement [23]. As shown in Figure 3a–c, the viscosity of the PSPs solutions was positively correlated with the concentration and Mw. This may be due to the increase in the solution concentration, leading to a corresponding increase in the intermolecular friction, which enhances the interaction between the polysaccharide molecules. The following is the order of the apparent viscosities of different PSPs at the same concentration: PSP-As > PSP-Us > PSP-Ms > PSP-Hs > PSP-Es > PSP-Fs. In contrast, the apparent viscosity of the PSPs was inversely related to the temperature of the system (Figure 3d–f). According to the results, an increase in temperature increases the volumetric expansion of the solution and decreases the intermolecular interaction force, thereby decreasing the apparent viscosity of the PSPs.

#### 3.1.8. Frequency Sweep Measurements

High-frequency modulus behavior could illustrate polysaccharide chain properties according to Edwards’ theory [40,41]. As shown in Figure 4, when the G′ and G″ curves of the dynamic flow curve of PSPs intersect, the solution state of the polysaccharides begins to change from liquid to gel-like, and its internal molecular chains are coupled and entangled, forming a reticular structure [42]. When G′ > G″, the sample solution is mainly solid elastic. Based on the above results, PSPs have a wide application prospect in the thickening process as well as the gel process.

#### 3.1.9. Antioxidant Activity of PSPs

The excessive production of ROS in the epidermis can disrupt the homeostasis of the antioxidant defense system [43]. Therefore, it is necessary to develop natural antioxidants to protect the body from injury. The chemical composition, molecular structure, and degree of active extraction of polysaccharides are the main factors affecting the antioxidant activity of PSPs. Therefore, we further investigated the antioxidant activity of PSPs extracted by different methods. As shown in Figure 5a, the scavenging ability of PSPs on DPPH free radicals was as follows: Vc (IC_50_ = 0.078 mg/mL) > PSP-Es (IC_50_ = 8.456 mg/mL) > PSP-Ms > PSP-Fs > PSP-Hs > PSP-Us > PSP-As. As shown in Figure 5b, the OH-radical-scavenging activity of PSPs increased in a dose-dependent manner. The scavenging effect was as follows: Vc (IC_50_ = 0.019 mg/mL) > PSP-Hs (IC_50_ = 5.563 mg/mL) > PSP-Fs > PSP-Us > PSP-As > PSP-Ms > PSP-Es. As shown in Figure 5c, the chelating ability of all PSPs for Fe^2+^ was as follows: PSP-As (97.82%) > PSP-Us > PSP-Fs > PSP-Hs > PSP-Ms > PSP-Es.

The biological activity of polysaccharides mainly depends on their monosaccharide composition, glycosidic bond type, Mw, etc. [44]. It is generally believed that polysaccharides with a lower Mw have a better scavenging ability for DPPH radicals. At the same time, their antioxidant activity is also inseparable from their monosaccharide composition. According to the above results, PSP-Es and PSP-Ms, which do not contain uronic acid and have a moderate Mw, have a better DPPH-radical-scavenging ability. However, PSP-As, which contain uronic acid and have a high Mw, have the worst ability to scavenge DPPH radicals. Based on the experimental results, it was found that the polysaccharides with better OH-radical-scavenging activity were usually PSP-Hs and PSP-Fs, which contain uronic acid and have an intermediate Mw. Based on the results of the Fe^2+^-chelating ability of PSPs, the activity of PSP-As and PSP-Us is best due to the fact that they have the largest Mw.

#### 3.1.10. Hypolipidemic Activity of PSPs

Studies have found that many hyperlipidemias have been reported to be associated with the excessive intake of fat and cholesterol, including atherosclerosis, cardiovascular disease, diabetes, and so on [45]. PSPs have potential hyperlipidemic activity, so the effect of the extraction method on their activity was further explored. Figure 5d shows the in vitro inhibition of pancreatic lipase activity by PSPs prepared by different methods. The inhibitory capacity was as follows (8 mg/mL): PSP-Es > PSP-Us > PSP-Ms > PSP-Hs > PSP-Fs > PSP-As. Polysaccharides can regulate hyperlipidemia by inhibiting the activity of pancreatic lipase. In the present study, PSPs could inhibit lipase activity to a certain extent in a concentration-dependent manner, possibly through a combination of hydrophobic and electrostatic interactions, leading to a change in the conformation of the lipase, or they could absorb and encapsulate the lipase [46]. The PSP-Es pancreatic lipase showed the highest inhibition efficiency, indicating that EAE was an effective method for extracting polysaccharides with a high biological activity.

Figure 5e,f show the cholate-binding ability of PSPs prepared by different processes. The binding ability of PSPs to cholate increased in a dose-dependent manner. Among them, PSP-Fs had the best cholate-binding ability, with a glycolate binding rate of 36.18% and a taurocholate binding rate of 45.12%. The in vitro binding ability of PSPs extracted by different methods changed, which was consistent with the results of other studies [7]. The hydrophilicity of PSP-Es and PSP-Fs was affected by their Mw. A higher Mw thereby increased the surface aggregation, thus increasing the amount of binding to fats and cholesterol [47]. At present, some synthetic drugs have common side effects. Therefore, there is an urgent need for natural non-toxic drugs to treat hyperlipidemia [48]. In conclusion, PSPs have the potential to prevent hyperlipidemia, and EAE and FAE are effective treatments.

#### 3.1.11. Immunomodulatory Activity of PSPs

Figure 6 shows that none of the PSPs prepared by different processes were cytotoxic. PSP-Hs, PSP-As, PSP-Us, PSP-Ms, and PSP-Fs (10~100 μg/mL) and PSP-Es (10~250 μg/mL) could significantly promote the proliferation of RAW264.7 macrophages (*p* < 0.05). When the concentrations of PSP-Hs and PSP-Es were 100 μg/mL and the concentrations of PSP-As, PSP-Us, PSP-Ms, and PSP-Fs were 10 μg/mL, the proliferation rate of RAW264.7 macrophages was the highest (*p* < 0.001). Compared with PSP-Hs and PSP-Es, the monosaccharide composition of the other four PSPs was different. Therefore, it is speculated that the monosaccharide components of GalA, Gal, and Xyl may be able to promote cell proliferation.

NO can activate macrophages to kill pathogenic bacteria and tumor cells and improve immunity [49,50]. Figure 7 shows that PSP-Hs (100~500 μg/mL) can significantly promote NO secretion (*p* < 0.05). PSP-As, PSP-Us, PSP-Ms, and PSP-Fs (10~500 μg/mL) can also significantly promote the secretion of NO (*p* < 0.001). According to the above experimental results, PSP-Es had little effect on the secretion of NO in RAW264.7 cells, and the content of mannose (Man) was low combined with the composition of monosaccharides. However, the PSPs obtained by the other extraction processes can promote the secretion of NO in RAW264.7 cells, which is presumably related to the content of Man. The results showed that PSPs have good immunomodulatory activity.

## 4. Conclusions

In this work, six different preparation processes, HWE, AAE, UAE, EAE, MAE, and FAE, were used to extract PSPs. The extraction rate of PSP-Hs was the highest (14.86%). The results showed that the chemical composition, the surface morphology, the viscosity, and the antioxidant, hypolipidemic, and immunomodulatory activities of the six PSPs were different. However, they all share similar functional groups, thermal stability, and glycosidic bond compositions. PSP-Us and PSP-Fs proved that the high energy of an ultrasonic wave and the low temperature of −80 °C could split the PSPs into small fragments. In addition, PSP-Hs (IC_50_ = 5.563 mg/mL) had good OH-radical-scavenging activity. PSP-As had a better Fe^2+^-chelating ability, and PSP-Es (IC_50_ = 8.456 mg/mL) had a better DPPH-radical-scavenging ability. Among them, PSP-Fs had the best binding ability to cholate and better hypolipidemic activity. PSP-Es showed the strongest inhibitory effect on pancreatic lipase. Based on these results, it is speculated that the in vitro binding ability of polysaccharides is closely related to their apparent viscosity and relative molecular mass. Their immunomodulatory activity combined with the analysis of monosaccharide composition suggests that Man plays an important role in immunomodulatory activities. PSPs have good antioxidant, hypolipidemic, and immunomodulatory activities and rheological behavior. However, the higher structures of PSPs and their structure–activity relationships need to be further explored. In this study, PSPs prepared by different processes were systematically explored and their structure–activity relationships were further analyzed. According to the results, HWE can be used to meet the requirements of production, which is simple, economical, and has the highest extraction rate. The appropriate preparation process can be selected according to the required antioxidant activity index. PSP-Es or PSP-Fs can be used to achieve the purpose of lipid regulation. On the contrary, PSP-Es could not be used for immunoregulation. The above results provide some data support for the application of PSPs.

## Figures and Tables

**Figure 1 foods-12-02088-f001:**
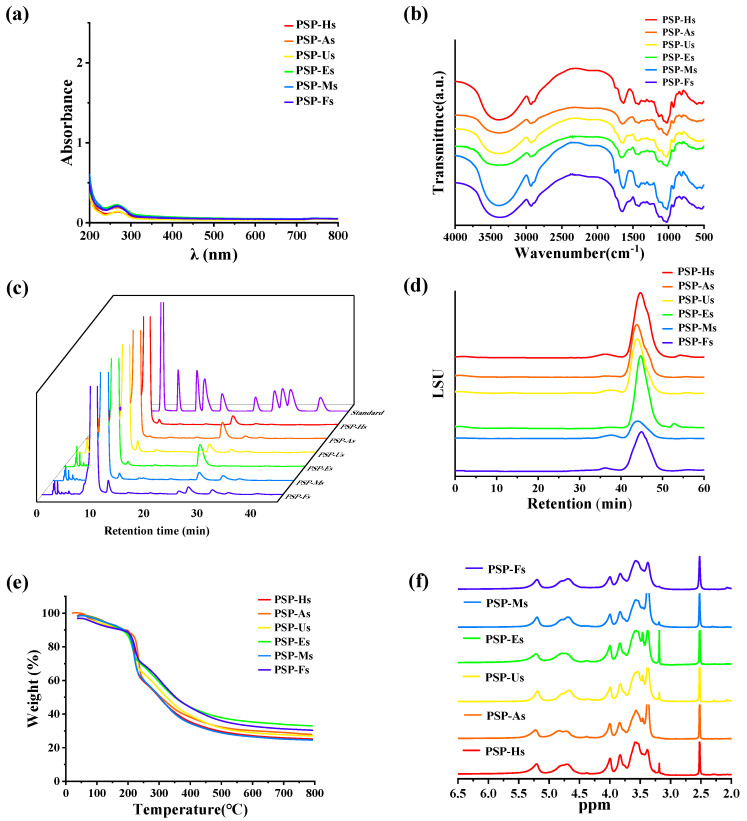
Characterization of PSPs: UV-vis spectra of PSPs (**a**); FT-IR spectra of PSPs (**b**); HPLCs of PSPs (**c**); HPGPCs of PSPs (**d**); TGAs of PSPs (**e**); and ^1^H-NMR chromatograms of PSPs (**f**).

**Figure 2 foods-12-02088-f002:**
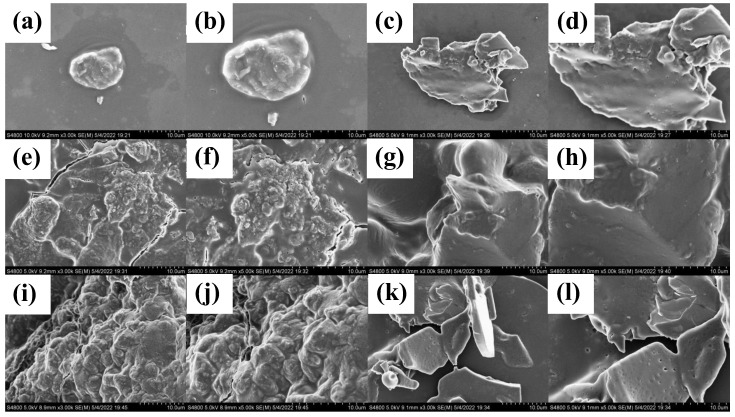
SEM image of PSPs (×3000 and ×5000): PSP-Hs (**a**,**b**); PSP-As (**c**,**d**); PSP-Us (**e**,**f**); PSP-Es (**g**,**h**); PSP-Ms (**i**,**j**); and PSP-Fs (**k**,**l**).

**Figure 3 foods-12-02088-f003:**
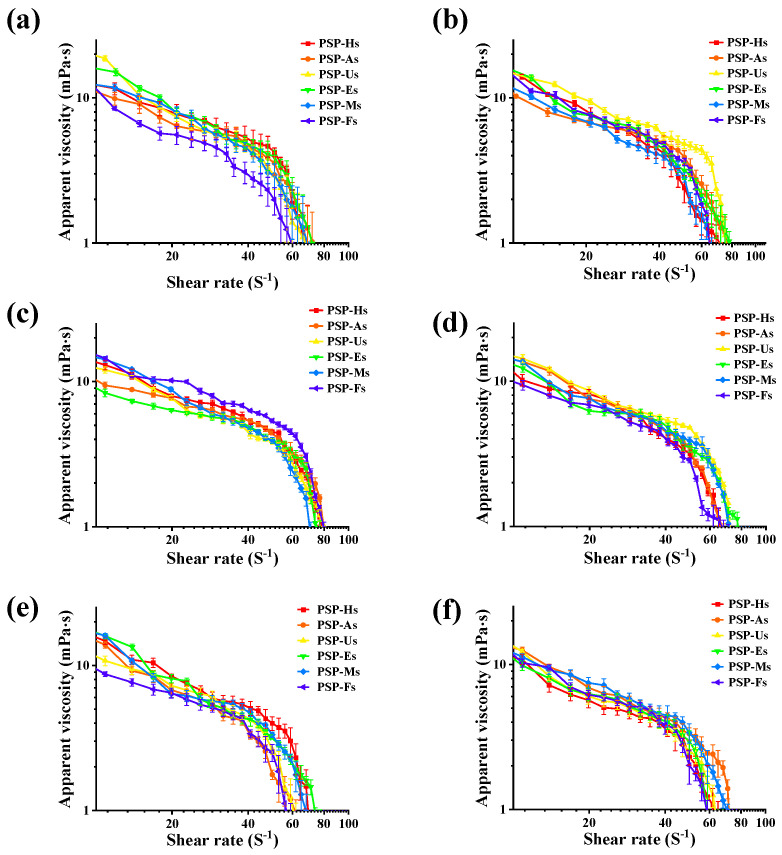
Apparent viscosity of PSPs at different concentrations: 4 mg/mL (**a**); 8 mg/mL (**b**); and 16 mg/mL (**c**) at a temperature of 25 °C. Apparent viscosity of PSPs at different temperatures: 20 °C (**d**); 60 °C (**e**); and 100 °C (**f**) at a concentration of 4 mg/mL.

**Figure 4 foods-12-02088-f004:**
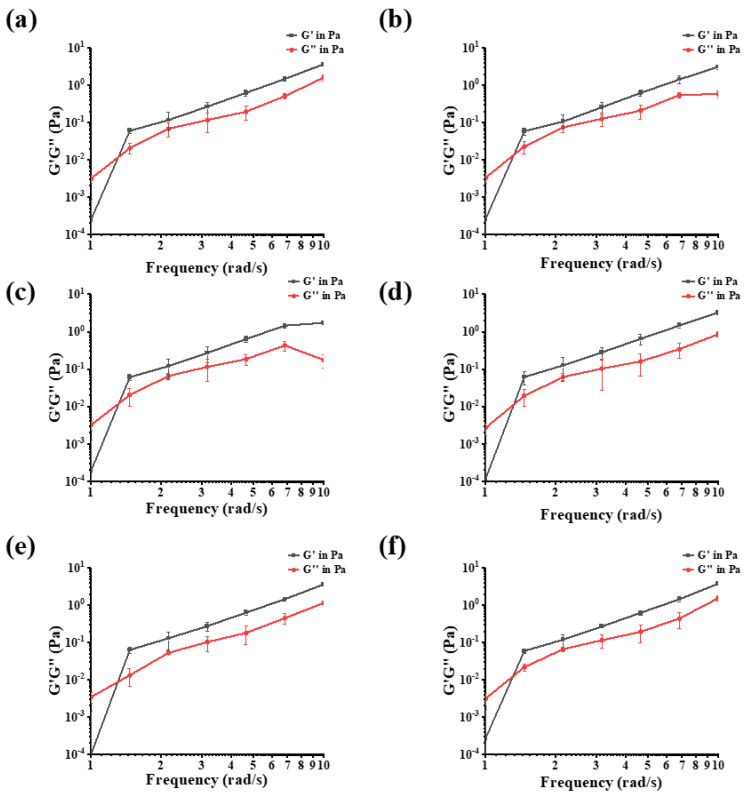
The viscoelastic properties of dynamic frequency sweep measurements for 4 mg/mL solutions: PSP-Hs (**a**); PSP-As (**b**); PSP-Us (**c**); PSP-Es (**d**); PSP-Ms (**e**); and PSP-Fs (**f**) at a temperature of 20 °C.

**Figure 5 foods-12-02088-f005:**
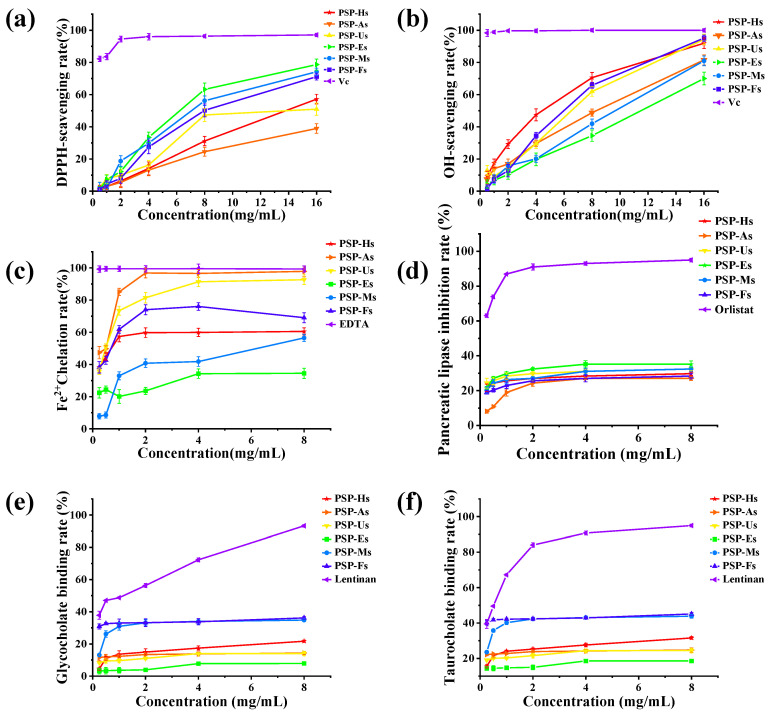
Antioxidative activity of PSPs: DPPH-radical-scavenging ability (**a**); OH-radical-scavenging ability (**b**); and Fe^2+^-chelating ability (**c**). The hypolipidemic activity of PSPs: pancreatic lipase inhibitory ability (**d**); glycocholate binding capacity (**e**); and taurocholate binding capacity (**f**).

**Figure 6 foods-12-02088-f006:**
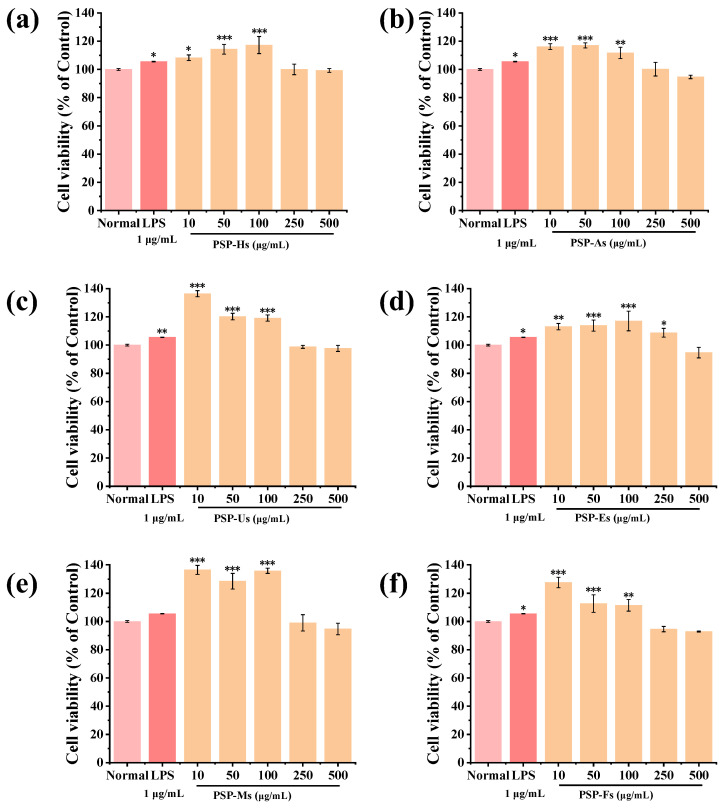
Effects of PSPs on phagocytosis of RAW264.7 cells: PSP-Hs (**a**); PSP-As (**b**); PSP-Us (**c**); PSP-Es (**d**); PSP-Ms (**e**); and PSP-Fs (**f**). (Note: * *p* < 0.05, ** *p* < 0.01, and *** *p* < 0.001 vs. control group).

**Figure 7 foods-12-02088-f007:**
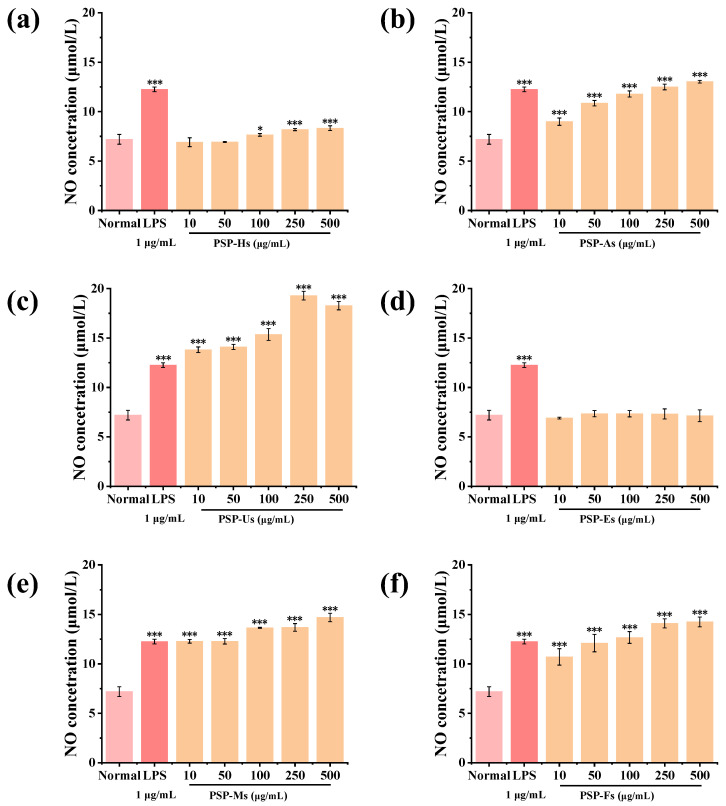
Effects of PSPs on NO secretion in RAW264.7 cells: PSP-Hs (**a**); PSP-As (**b**); PSP-Us (**c**); PSP-Es (**d**); PSP-Ms (**e**); and PSP-Fs (**f**). (Note: * *p* < 0.05 and *** *p* < 0.001 vs. control group).

**Table 1 foods-12-02088-t001:** Extraction rate and total soluble sugar content of PSPs.

Sample	Extraction Rate (%)	Total Soluble Sugar Content (%)
PSP-Hs	14.86 ± 3.14 ^a^	80.07 ± 2.70 ^b^
PSP-As	9.62 ± 2.02 ^bc^	78.10 ± 4.33 ^b^
PSP-Us	5.83 ± 2.52 ^cd^	76.61 ± 1.74 ^b^
PSP-Es	12.00 ± 3.26 ^ab^	84.64 ± 2.11 ^a^
PSP-Ms	6.71 ± 1.49 ^cd^	87.14 ± 1.69 ^a^
PSP-Fs	3.40 ± 1.01 ^d^	77.52 ± 1.28 ^b^

Different lowercase letters in the same column indicate a statistically significant difference (*p* < 0.05).

**Table 2 foods-12-02088-t002:** FT-IR analysis results for PSPs.

Absorption Peak/cm^−1^	Functional Group
3276	-OH stretching vibration absorption peak
2933	C-H stretching vibration of CH_2_ group
1743	C=O vibration absorption peak
1629	Vibrational absorption peaks induced by O-H/C-O/COO-
1414	C-H, CH_2_ bending vibration absorption peaks
1126	Variation-angle vibration peaks of alcohol hydroxyl groups
1014/929	Presence of pyranose
875	Characteristic absorption peaks of β-pyranose
875/813	Presence of mannose (Man)

**Table 3 foods-12-02088-t003:** The molar percentage of the monosaccharide composition of PSPs.

Sample	Monosaccharide Composition (mol%)				
	Man	GalA	Glc	Gal	Xyl
PSP-Hs	0.212	0.086	0.559	0.143	-
PSP-As	0.190	0.039	0.768	0.122	0.052
PSP-Us	0.302	-	0.516	-	0.182
PSP-Es	0.050	-	0.950	-	-
PSP-Ms	0.165	-	0.502	-	0.333
PSP-Fs	0.309	0.136	0.378	0.178	-

- means not detected.

**Table 4 foods-12-02088-t004:** ^1^H-NMR chemical shift values of PSPs.

Sample	δ1	δ2	δ3	δ4	δ5	δ6	δ7	δ8	δ9	δ10
PSP-Hs	2.511	3.190	3.387	-	3.558	3.841	4.007	4.693	4.819	5.202
PSP-As	2.514	3.190	3.396	3.462	3.560	3.850	4.012	4.715	4.851	5.216
PSP-Us	2.511	3.188	3.388	3.463	3.561	3.840	4.009	4.681	4.792	5.207
PSP-Es	2.515	3.187	3.388	3.467	3.569	3.845	4.008	4.693	4.807	5.219
PSP-Ms	2.516	3.185	3.385	3.463	3.567	3.842	4.005	4.689	4.805	5.199
PSP-Fs	2.518	3.189	3.386	-	3.569	3.843	4.000	4.680	4.812	5.212

- means not detected.

## Data Availability

Data is contained within the article.

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
