# Peer review of "Effects of Extraction Methods on the Physicochemical Properties and Biological Activities of Polysaccharides from Polygonatum sibiricum"

_foods, 2023, doi:10.3390/foods12102088_

Round 1
Reviewer 1 Report
After reading the paper, I thought the discussion about the Biology of polysaccharides such as the antioxidant and immunomodulatory activities was too short. There were some part of the writing that I am not sure I understand. For example, lines 33-34: "Polygonatum was sweet in taste, flat in nature, belongs to the spleen, lung, and kidney meridians", I am not sure what the authors meant by saying "belongs to the spleen", and perhaps the authors could revise the whole sentence. Anyways, I have the following comments:
1. In the abstract, when the authors refer to PSP-A and the others, please give the full name at the first mention.
2. Lines 275-276, the authors wrapped things up by concluding that GalA and Gal may contribute to the antioxidant effect of the polysaccharides. I have to argue with that. I cannot see how GalA and Gal contribute to the antioxidant activity of the polysaccharides. For example, PSP-A, -H, and -F contain those two monosaccharides. If GalA and Gal contribute to such activity, then all 3 should have strong antioxidant activities in all assays. However, those 3 are among the worst in DPPH assay. In addition, PSP-U, which doesn't have those 2 monosaccharides, was among the best that has the chelating activity. Thus, the authors could not conclude with such statement. Please revise.
3. Line 276: "Consistent with the results of wang et al." is not a complete sentence.
4. Lines 322-323: "It was believed that GalA and Gal promote the secretion of NO factors." Same as #2. PSP-U doesn't have GalA and Gal but it is a strong inducer of NO. Please explain.
5. Please explain the NO production results from PSP-E, which did not produce NO at all.
6. Importantly, please provide research rationalization for why this work should be conducted.
minor revision on the writing is required.
Reviewer 2 Report
The manuscript was well written about the different methods in extraction of the polysaccharide of Polygonatum sibiricum. The following comments are provided to improve its citeability and quality:
The introduction and literature review for the polysaccharide is weak and should be improved. Furthermore, the aim of the work should be clearly determined and emphasized.
Line 55 is incomplete.
Line 66: it is necessary to mention all of the materials used in the work.
Line 70: the method is not clear, how did you dry? How did you crush?
Line 89-90: the specific characterization of enzyme did not determine?
All parts of the experiments must be explained with more details. It is not possible for readers to repeat the experiments.
Line 124-130: why the authors used only these concentrations for the rheological measurements. As mentioned earlier, the methods are not complete. For instance, what is the probe for the SAOS measurements and its gap?
Table 1: it is required to provide the alphabetical statistical significance.
SEM figures did not imply the correct action of the making photos. Please repeat the experiments.
For the FTIR measurements and rheological properties of the polysaccharide, I suggest seeing the works performed on Basil seed gum such as “Dynamic viscoelastic study on the gelation of basil seed gum” . It seems the work is required to compare with the similar works such as the papers particularly performed on the polysaccharide and with this comparison, the discussion did not approach to the high quality.
The manuscript was well written about the different methods in extraction of the polysaccharide of Polygonatum sibiricum. The following comments are provided to improve its citeability and quality:
The introduction and literature review for the polysaccharide is weak and should be improved. Furthermore, the aim of the work should be clearly determined and emphasized.
Line 55 is incomplete.
Line 66: it is necessary to mention all of the materials used in the work.
Line 70: the method is not clear, how did you dry? How did you crush?
Line 89-90: the specific characterization of enzyme did not determine?
All parts of the experiments must be explained with more details. It is not possible for readers to repeat the experiments.
Line 124-130: why the authors used only these concentrations for the rheological measurements. As mentioned earlier, the methods are not complete. For instance, what is the probe for the SAOS measurements and its gap?
Table 1: it is required to provide the alphabetical statistical significance.
SEM figures did not imply the correct action of the making photos. Please repeat the experiments.
For the FTIR measurements and rheological properties of the polysaccharide, I suggest seeing the works performed on Basil seed gum such as “Dynamic viscoelastic study on the gelation of basil seed gum” . It seems the work is required to compare with the similar works such as the papers particularly performed on the polysaccharide and with this comparison, the discussion did not approach to the high quality.
Reviewer 3 Report
I have read the manuscript and I have a few questions and recommendations. 1. Abbreviations must be deciphered in the first place of mention.
2. In section 2.2.5, it is necessary to indicate the activity of enzymes, the manufacturer of enzymes.
3. In section 2.2.4, please indicate the brand of the ultrasonic generator and its power.
4. In section 2.2.6, please indicate the brand of microwave extractor, manufacturer, its power and exposure mode.
5. In section 2.3.1, please indicate the solvent and polysaccharide concentration for the spectrum, as well as the model and manufacturer of the spectrophotometer.
6. How were the FTIR spectra obtained: from a solution or from a tablet with KBr?
7. Give in more detail the methods for determining sugar content and protein content, indicating the linearity of the method, the range of measured concentrations and literatury references for the authors of the methods.
8. For Analysis of monosaccharide composition, please provide chromatographic conditions, hydrolysis conditions, standard sample chromatograms, and chromatograms of all your polysaccharide samples. For section 2.3.2, provide relevant references.
9. In section 2.3.4, specify the sample preparation.
10. In section 2.4.1 Vc what is it? Specify the name of the reference substance, literature reference to its use in DPPH and OH scavenging rate. Give the value of IC 50 for Vc from the literature.
11. In section 2.4.2, indicate the range of linearity of the method and its R2.
12. Please supplement section 2.4.3 with the data of the commonly used reference substance with immunomodulatory activity. Provide a literary reference to the Griess method. Indicate the range of method linearity, its linearity. Why was only NO release determined?
13. In section 3.1.1, provide evidence that in Figure 1a, PSPs all contain a small amount of protein. Include a protein spectrum in addition to this figure.
14. The legend for figure 1 is not correct.
15. Please explain on what basis of your experimental data and literature data you concluded that "... the results showed that PSPs contained polysaccharides with β-ring glycosidic linkages".
16. In section 3.1.3, please provide the formula for calculating the monosaccharide composition.
17. Please provide relevant references to support your statement: "...The biological activity of polysaccharides mainly depends on their monosaccharide composition, glycosidic bond type and Mw ...". For sulfated polysaccharides of brown algae, it has been established that the molecular weight, composition, and activity depend on the technology and, accordingly, affect the activity (for example, https://doi.org/10.3390/md20100606). To confirm your conclusions, please conduct a correlation analysis of molecular weight and antioxidant activity. How do protein impurities affect? provide statistical analysis data.
18. How can you explain the different activity of PSPs on NO secretion of RAW264.7, especially the PSPs-E and PSPs-H samples. At the same time, these samples had an effect comparable to other samples on phagotosis.
19. In the introduction, there is no justification for the relevance of the study. Please improve this section.
Reviewer 4 Report
The introduction is short and is not giving background to why to look into these polysaccharides. Explanation of different terms as yin and qi is needed for the broader audience.
Material and methods; All chemicals used should be listed with grades and brand as it is now it is only for few. Ex. it is important to know the enzymes with activity. The sentences are short and more for a protocol and not in text format. For the different extraction methods more information is needed as it is unclear what is meant with continues as in xxx. Be more precise.
For the enzyme extraction boiling is used to inactivate the enzymes this will influence the extraction. Please comment on this. Dialysis could be another solution.
In vitro and similar words should be in italics.
Results: There is use of extraction rate, yield rate and yield for I think the same results. Please be consistent and use the same terme all through the manuscript.
There is a lot of data and some could be placed in supplementary and fokus on what is necessary. For all figure and table legends - the text should be expanded for the reader to be able to understand without the full text. This is not possible now. The monosaccharide composition is different in the different extractions. It seems that pectin is missing from some (no GalA) this could be commented. As the result part is also the analysis part it would be beneficial to have more analysis. From how the text is written it seems that the authors have mannose coming from cellulose - cellulose is made of glucose.
Fig 2; I am not convinces the pictures show polysaccharides as the fragments resemble cell wall fragments. This is valid in itself to see the influence of the different extraction methods.
Antioxidant activity - describe where the epidermis is.
It is not clear whether there is made statistic for figs 5, 4, 3.
The text in the methods part need revising and all text a more thorough read.
Reviewer 5 Report
Effects of extraction methods on the physicochemical properties and biological activities of polysaccharides from Polygonatum sibiricum
Line number 22: Which extraction method offered higher antioxidant activity?
Line number 26: Mention the main findings of the study
Line number 28: Re-check keywords
Line number 32: Elaborate on the background of Polygonatum sibiricum, its production statistics, etc.
Line number 41: Extraction method influenced extraction rates etc. How?
Line number 48. Please add these papers for ultrasound extractions:
An inclusive overview of advanced thermal and nonthermal extraction techniques for bioactive compounds in foods and food-related matrices. Food Reviews International, 38(6), 1166-1196.
Sonication, a Potential Technique for Extraction of Phytoconstituents: A systematic review. Processes, 9(8), 1406.
Line number 50-55: Re-check the phrase, and the paragraph
Line number 120: DMSO: Mention the name of the reagent
Line number 61. Associate. Is it necessary to mention the name of the teacher?
Line number 89. name should be italic.
Line number 94. name should be italic.
Line number 126: Re-check the shear rates
Line number 132: Re-check the method
Line number 137: Grammar issues in method 2.4.2.
Line number 168.169: Repetition of words
Line number 171, 184, 198: Mention citation for this finding
Line number 198: Add some cross-references
Line number 201: Add some references to validate your statement and elaborate on these findings
Line number 204, 225: Add some previous studies
Line number 255: Add some cross-references
Line number 306: Add some cross-studies and re-check the range and units mentioned
Line number 349: Mention which extraction method offered the best result in terms of all biological activities or suggest which can be used in the future for better results/ further applications
Line number 409, 414, 420, 428, 438, 451: Page numbers are missing
Line number 436, 438, 458: Volume number is missing
fair
Round 2
Reviewer 2 Report
All the comments are replied properly. Please be considered that the 3.1.2. Functional group composition of PSPs is so critical in the properties of the polysaccharide and it is recommended to compare with the recent works on the functional groups of other polysaccharides such as Basil seed gum. Therefore, please consider this work "Effect of thermal treatment on chemical structure of β-lactoglobulin and basil seed gum mixture at different states by ATR-FTIR spectroscopy”. It is important to distinguish the differences in the functional groups of the polysaccharide with the similar ones.
Reviewer 4 Report
Thank you for the revised manuscript and many of the issues are now improved.
A few issues is still needed to be addressed.
Especially in the material and methods.
Please indicate the time and speed for the centrifugation.
Please indicate the filter conditions - as size and type of filter.
Thank you for the addition of chemicals and brand names. If the company where the cellulase is bought only have this type it is OK not to mention the catalogue number.
I can notice that Alpha-amylase, Chloroform and Acarbose all are mentioned as chemicals in the list but I can not see them later in the methods. Is this a mistake?
For the statistics is mentioned all data are shown with +/- SD - this can not be seen. You mention in the comments that for fig 3,4 and 5 this is due to no difference in the data sets. Please make a comment of this.
Line 263 - in one of the sentences - Gal and Gal is mentioned - is this a mistake?
The material and method part is improved but could benefit from an extra write through.
